# Late Inguinal Swelling: Don’t Judge a Book by Its Cover! An Unusual Case of Lymphocele

**DOI:** 10.3390/reports7010020

**Published:** 2024-03-11

**Authors:** Francesco Natale, Giovanni Cimmino

**Affiliations:** 1Vanvitelli Cardiology Unit, Monaldi Hospital, 80131 Naples, Italy; natalefrancesco@hotmail.com; 2Cardiology Unit, Azienda Ospedaliera Universitaria Luigi Vanvitelli, 80138 Naples, Italy; 3Department of Translational Medical Sciences, Section of Cardiology, University of Campania Luigi Vanvitelli, 80131 Naples, Italy

**Keywords:** swelling, vascular closure device, ultrasound, lymphocele

## Abstract

A 58-year-old man with a history of diabetes type I and chronic coronary syndrome who underwent coronary artery bypass grafting (CABG) 7 years before was admitted to a cardiology unit for unplanned cardiac catheterization because of dyspnea and chest pain at rest. Femoral access was chosen because of the previous CABG and a vascular closure device (VCD) was used at the end of the procedure. Because of femoral artery rupture during VCD implantation, surgical vascular repair was performed. About 45 days later, the patient experienced a growing inguinal swelling at the site of vascular access in the absence of fever and clinical features of inflammation. The swelling became painful over time. Despite the most probable hypothesis of a hematoma, pseudoaneurysm, and inguinal abscess, a final diagnosis of lymphocele was made.

## 1. Introduction

The percutaneous treatment of coronary artery disease (CAD) [1] and some structural heart diseases [2] is evolving and expanding in clinical practice. More and more patients with different types of heart diseases are being treated by percutaneous or transcatheter interventions. Radial and femoral accesses are usually used [3]. However, vascular access complications may occur. Based on the types of complications, different treatment options might be available [4,5,6]. Minor complications include minor bleeding, ecchymosis, and stable hematoma. Major complications include pseudoaneurysm, arteriovenous (AV) fistula, hematoma requiring transfusion, retroperitoneal hemorrhage (for femoral access), arterial dissection, embolism, thrombosis, infection, and vessel rupture/perforation [7]. A less common complication is lymphocele. It is a collection of lymphatic fluid that usually occur after pelvic surgery [8]. However, potentially any part of the body where lymphatic tissue is resected (e.g., lymphadenectomy) or injured in trauma might be affected by lymphocele formation. Thus, it should be taken into account in the diagnostic work-up of any swelling.

## 2. Detailed Case Description

A 58-year-old man with a history of diabetes type I and chronic coronary syndrome who had undergone coronary artery bypass grafting (CABG) 7 years before with left internal mammalian artery (LIMA) on left descending artery, right internal mammalian artery (RIMA) on circumflex coronary artery, and saphenous venous graft on right coronary artery, was admitted to the cardiology unit for unplanned cardiac catheterization because of dyspnea and chest pain at rest. As is our practice in cases of the coexistence of LIMA and RIMA grafts, femoral access was chosen because of the previous CABG and a vascular closure device (VCD) was used at the end of the procedure. Because of femoral artery rupture during VCD implantation, surgical vascular repair was performed. In brief, a longitudinal incision, starting approximately 8–10 mm above the inguinal ligament and going on downward, was used to expose the femoral artery. It was repaired using a continuous suturing technique with a 6-0 polypropylene suture. The procedure was performed as for standard practice without complications. 

About 45 days after, the patient experienced a growing inguinal painful swelling at the site of vascular access. The patient was evaluated in our outpatient office. Upon examination, a large painful swelling in the right inguinal region was detected without clinical features of inflammation (no redness and heat; white arrow, Figure 1A,B). No fever was reported. An ultrasound evaluation was performed showing an echo-free lesion (Figure 1C) with a vascular pedicle well defined by the color Doppler with no flow communication with the lumen (white arrow, Figure 1D). We performed a fine needle aspiration removing more than 20 mL of citrine yellow liquid from the swelling (white arrow, Figure 1E,F). Taking into account the absence of fever, the type of aspirated fluid and the recent inguinal surgery, a diagnosis of lymphocele [9] was made as the primary hypothesis. It was later confirmed by the biochemical analysis of the liquid. A compression bandage was applied for two weeks as per the surgeon’s indication. A follow-up visit at one and two months after the aspiration was scheduled. The lymphocele did not recur after the aspiration.

Other possible hypotheses were hematoma, pseudoaneurysm, and inguinal abscess, which are usually associated with the type of procedure that the patient underwent [7]. Lymphocele is usually a surgical complication and is more often asymptomatic [9]. It usually develops when the lymphatic drainage gets damaged during surgery, resulting in lymph fluid that drains out from the lymphatic channel with cavity formation nearby. Larger lymphoceles may cause symptoms mainly related to the compression of adjacent structures [10]. Smaller lymphoceles may regress spontaneously over time but if symptomatic, needle aspiration, catheter insertion and drainage, and surgical drainage are indicated [9,10]. In our case, fine needle aspiration was therapeutic.

## 3. Discussion

This is a case of post-surgical lymphocele that occurred after repair of the femoral artery used as vascular access for coronary angiography. The site is uncommon as most lymphoceles occur after extensive pelvic surgery [8]. Upon inspection of the inguinal region, an infective complication of the previous surgery was the primary hypothesis. However, the absence of any clinical features of inflammation, such as redness and heat, and no fever since the surgery were the key elements for an alternative hypothesis. The image is of educational importance since a diagnosis of lymphocele should be considered in the presence of swelling like the one reported here. Lymphocele is a postsurgical and/or post-traumatic complication that develops when the lymphatic system gets damaged. Because of this damage, the lymph fluid drains out from the lymphatic ducts, building up in a cavity nearby. Small lymphoceles resolve spontaneously, while large lymphoceles, if not treated in time, might obstruct blood flow to the treated site and compress the surrounding blood vessels, thus delaying wound healing and increasing the risk of infection.

## Figures and Tables

**Figure 1 reports-07-00020-f001:**
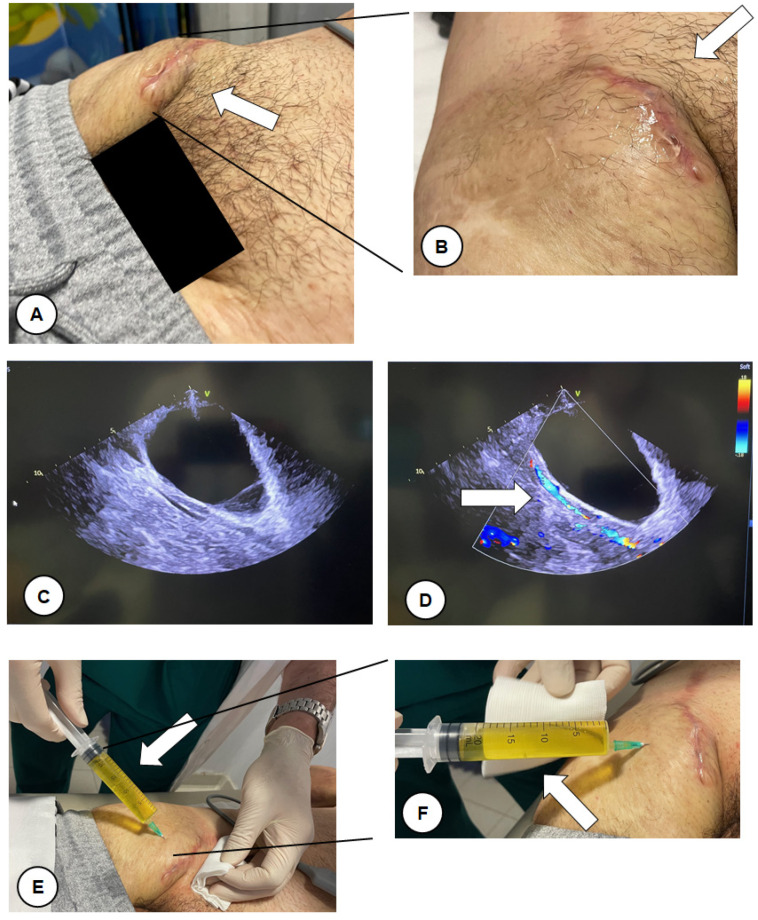
Large swelling in right inguinal region without clinical features of inflammation (no redness and heat; white arrow (**A**,**B**)). No fever was reported. The ultrasound evaluation showed an echo-free lesion (**C**) with a vascular pedicle well defined by the color Doppler with no flow communication with the lumen (white arrow (**D**)). A fine needle aspiration was performed, removing more than 20 mL of citrine yellow liquid from the swelling (white arrow (**E**,**F**)).

## Data Availability

The data underlying this study are available in this article.

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
