# Peer review of "Late Inguinal Swelling: Don’t Judge a Book by Its Cover! An Unusual Case of Lymphocele"

_reports, 2024, doi:10.3390/reports7010020_

Round 1

Reviewer 1 Report

Comments and Suggestions for Authors

The manuscript is interesting, but I have some comments and questions

for the authors:

1.     Page 1 line 14; line 34: Specify why the previous CABG was the reason why you chose the femoral access for cardiac catheterization?

2.     Page 2 line 36: Exactly which vascular repair was performed ? Were there any difficulties during the approach to the femoral artery? 

3.     There is no information about the follow-up. Has the lymphocele recurred after aspiration? How long was the observation period? Did you apply compression after aspiration??

4.     I think the figure 1 A and E are redundant. 

Author Response

The manuscript is interesting, but I have some comments and questions for the authors:

ANSWER: We thank the reviewer for his/her kind word on our "case"

  1. Page 1 line 14; line 34: Specify why the previous CABG was the reason why you chose the femoral access for cardiac catheterization?

ANSWER: We thank the reviewer for his/her question. Patient underwent to triple CABG with left internal mammalian artery (LIMA) on left descending artery (LAD), right internal mammalian artery (RIMA) on circumflex coronary artery and saphenous venous graft (SVG) on right coronary artery. It our practice the use of femoral access in case of coexistence of LIMA and RIMA grafts

  1. Page 2 line 36: Exactly which vascular repair was performed ? Were there any difficulties during the approach to the femoral artery?

ANSWER: We thank the reviewer to point this out. A longitudinal incision, starting approximately 8-10 mm above the inguinal ligament and going on downward, was used to expose the femoral artery that was repaired using continuous suturing technique with a 6-0 polypropilene suture. The procedure was performed as for standard practice without complications

  1. There is no information about the follow-up. Has the lymphocele recurred after aspiration? How long was the observation period? Did you apply compression after aspiration??

ANSWER: We apologize for the missing information. Lymphocele did no recur after aspiration. A compression bandage was applied for two weeks as for surgeon indication. A follow up visit at one and two months after aspiration was scheduled

  1. I think the figure 1 A and E are redundant.

ANSWER: We thank the reviewer for his/her suggestion. However, we would like to keep Figure 1 A as demonstration of lesion appearance at first clinical examination and Figure 1 E as demonstration of fine needle aspiration

Reviewer 2 Report

Comments and Suggestions for Authors

Dear Authors,
You presented an interesting case of lymphocele as a complication after endovascular femoral access. First of all, I would suggest an extensive language check-up, as in some parts of the article, language is too colloquial and commonly grammatically incorrect. 

Second of all I would suggest removing the introduction and adding some discussion at the end of the article. 

Last but not least, the follow-up is missing. What was the future outcome in this case? If you have any further information on this patient (even the short-term control) please add it to the article.

Kind regards

Comments on the Quality of English Language

I would suggest an extensive language check-up, as in some parts of the article, language is too colloquial and commonly grammatically incorrect. 

Author Response

You presented an interesting case of lymphocele as a complication after endovascular femoral access.

We thank the reviewer for his/her kind words on our "interesting Image"

First of all, I would suggest an extensive language check-up, as in some parts of the article, language is too colloquial and commonly grammatically incorrect.

ANSWER: We apologize for typos and grammar errors. Article has been extensively edited

Second of all I would suggest removing the introduction and adding some discussion at the end of the article.

ANSWER: We thank the reviewer for his/her suggestion. We added some discussion.

Last but not least, the follow-up is missing. What was the future outcome in this case? If you have any further information on this patient (even the short-term control) please add it to the article.

ANSWER: We apologize for the missing information. Follow-up has been added.

Reviewer 3 Report

Comments and Suggestions for Authors

I have read with great attention the case report titled "Late inguinal swelling: don't judge a book by its cover!" Honestly, though, I don't understand why such a common, straightforward, and routine clinical case should be considered worthy of publication in any way.

While convinced of the excellent clinical work of the authors, I find no element in this paper that makes it "useful" to the international scientific literature. That being said, I would still like to suggest to the authors to organize their paper into sections, as is commonly requested: introduction, case report, discussion, and conclusions.

Author Response

I have read with great attention the case report titled "Late inguinal swelling: don't judge a book by its cover!" Honestly, though, I don't understand why such a common, straightforward, and routine clinical case should be considered worthy of publication in any way.

While convinced of the excellent clinical work of the authors, I find no element in this paper that makes it "useful" to the international scientific literature.

ANSWER: With the respect for the reviewer, we believe that these images are really impressive and of educational importance for the young doctors because the presence of a swelling after vascular procedure could lead a primary diagnosis of haematoma, but other complications should be considered. It si important for young doctors to draw the attention on the clinical features of any lesion, since the absence of both, fever and clinical features of inflammation, should be of help to make a right diagnosis. This is the reason why, to better highlight this concept, we have chosen the proposed title

That being said, I would still like to suggest to the authors to organize their paper into sections, as is commonly requested: introduction, case report, discussion, and conclusions.

ANSWER: We thank the reviewer for his/her suggestion. We have reorganized this paper.

Round 2

Reviewer 1 Report

Comments and Suggestions for Authors

I have one more question for the authors. Explain please the probable mechanism of lymphocyte development. What do you think? What could be its reason?

Author Response

We thank the reviewer to point this out.

As for the majory of lymphocele, it usually develops when the lymphatic drainage gets damaged during surgery, resulting in lymph fluid that drains out from the lymphatic channel with a cavity formation nearby

We have added this sentence to the revised version of the maniscript

Reviewer 2 Report

Comments and Suggestions for Authors

Dear Authors,

Thank you for editing and improving the article. At this point I do not have any further reuqests or suggestions.

Kind regards

Author Response

We thak the reviewer

Reviewer 3 Report

Comments and Suggestions for Authors

I confirm my previous comments 

Comments on the Quality of English Language

I confirm my previous comments 

Author Response

We respect the reviewer comment

Round 3

Reviewer 1 Report

Comments and Suggestions for Authors

I have no more questions for the authors